# Clinical Significance of B-Type Natriuretic Peptide and N-Terminal Pro-B-Type Natriuretic Peptide in Pediatric Patients: Insights into Their Utility in the Presence or Absence of Pre-Existing Heart Conditions

**DOI:** 10.3390/ijms25168781

**Published:** 2024-08-12

**Authors:** Kamila Maria Ludwikowska, Monika Tokarczyk, Bartłomiej Paleczny, Paweł Tracewski, Leszek Szenborn, Jacek Kusa

**Affiliations:** 1Department of Pediatric Infectious Diseases, Wroclaw Medical University, Ludwika Pasteura 1, 50-367 Wrocław, Poland; kamila.ludwikowska@umw.edu.pl (K.M.L.); leszek.szenborn@umw.wroc.pl (L.S.); 2Department of Physiology and Pathophysiology, Wroclaw Medical University, Ludwika Pasteura 1, 50-367 Wrocław, Poland; bartlomiej.paleczny@umw.edu.pl; 3Department of Pediatric Cardiology, Regional Specialist Hospital in Wroclaw, Research and Development Center, Kamieńskiego 73a, 51-124 Wrocław, Poland; pawel.tracewski@gmail.com (P.T.); jacek.kusa@wssk.wroc.pl (J.K.); 4Pediatric Cardiology Department, Medical University of Silesia, Medyków 16, 40-752 Katowice, Poland

**Keywords:** heart failure, MIS-C, BNP, Kawasaki disease, children

## Abstract

The clinical significance of B-type natriuretic peptide (BNP) and N-terminal pro-B-type natriuretic peptide (NT-proBNP) in pediatric patients remains an area of evolving understanding, particularly regarding their utility in the presence or absence of pre-existing heart conditions. While clear cutoff values and established roles in heart failure are understood in adult patients, pediatric norms vary with age, complicating interpretation. Notably, the emergence of multi-system inflammatory syndrome in children (MIS-C) has highlighted the importance of these markers not only in the detection of acute heart failure but also as a marker of disease severity and even as a differential diagnosis tool. This review summarizes current knowledge on the utility of BNP and NT-proBNP in pediatric patients. Their unique physiology, including circulation and compensation mechanisms, likely influence BNP and NT-proBNP release, potentially even in non-heart failure states. Factors such as dynamic volemic changes accompanying inflammatory diseases in children may contribute. Thus, understanding the nuanced roles of BNP and NT-proBNP in pediatric populations is crucial for the accurate diagnosis, management, and differentiation of cardiac and non-cardiac conditions.

## 1. Introduction

Natriuretic peptides (NPs)—the brain natriuretic peptide (BNP) and its N-terminal prohormone (NT-proBNP)—play crucial roles in regulating cardiovascular homeostasis. Elevated levels of these peptides in the blood indicate activated compensatory mechanisms that precede symptoms of heart failure. The significance of natriuretic peptide levels in heart failure is well established [1]. However, in pediatric populations, differences in cardiovascular physiology and the epidemiology of heart failure, including underlying causes, exist. Interpreting NP levels in children can be more complex compared to adults.

For children with chronic heart diseases, NPs serve as valuable biomarkers aiding in diagnosis and monitoring. Released by the heart in response to increased pressure or volume overload, their levels help assess cardiac function, particularly exacerbations of chronic heart failure. Yet, interpreting BNP or NT-proBNP levels in children presents challenges due to factors such as age, gender, and comorbidities. Unlike adults, age-specific reference ranges and guidelines for NPs testing and interpretation in pediatric patients lack solid scientific evidence.

Although NPs are frequently used in diagnosing and monitoring heart conditions in children, their levels may rise in acute illnesses even without underlying heart issues. Since the coronavirus disease 2019 (COVID-19) pandemic and the emergence of multi-system inflammatory syndrome in children (MIS-C), BNP and NT-proBNP have seen increased utilization as laboratory tests in acute inflammatory disorders among children. The recognition of MIS-C has underscored the importance of these biomarkers in assessing cardiac involvement and monitoring the course of inflammatory conditions in pediatric patients. Their role extends beyond traditional heart failure scenarios, highlighting their relevance in the broader spectrum of pediatric inflammatory diseases. Thus, there is a need to better understand the current state of knowledge on NPs in children.

In conditions like sepsis, respiratory infections, or dehydration, NP levels can increase due to physiological stress on the body. These conditions induce volume overload, inflammation, and hemodynamic changes, prompting BNP and NT-proBNP release from the heart as a compensatory mechanism to regulate fluid balance and maintain cardiac function. Children, having a larger proportion of water in body compartments, are more susceptible to dynamic changes in NP levels than adults, even in the absence of underlying heart problems. Consequently, during feverish illnesses, elevated BNP or NT-proBNP levels may indicate the body’s response to acute illness rather than a primary cardiac concern.

In our review, we will discuss the current understanding of BNP and NT-proBNP utility in children, taking into account clinical contexts such as fever, infection, or inflammation.

## 2. Methodology

In this manuscript, we analyzed numerous publications based on their usefulness in evaluating NT-proBNP and BNP concentrations in the pediatric population. MEDLINEand Google Scholar databases were searched using the following terms:

*BNP or NT-proBNP or NP or heart enzymes or natriuretic peptides or NPs or brain natriuretic peptide and children or infants or neonates or adolescents or pediatric.* Additionally, for certain sections of the manuscript, we also searched the database for terms like: *inflammation, heart defects, cardiomyopathy, physiology, myocarditis, Kawasaki disease, multi-system inflammatory syndrome in children, MIS-C, sepsis.*

The primary list of chosen publications was proposed by K.M.L., M.T., B.P., P.T., and approved by all the authors.

If no publications for pediatric patients were available, we used publications regarding adult patients.

Studies from the last 5 years were included. If no data were available, the timespan was gradually expanded beyond 5 years.

## 3. Physiology of BNP and NT-proBNP

### 3.1. Overview of Physiological Significance, Biosynthesis, and Receptors

Since the discovery of “an extremely powerful” natriuretic substance in rat atrial myocytes in 1981 [2], two major cardiac NPs—atrial natriuretic peptide (ANP) and BNP—have emerged as critical regulators of the physiological response to cardiac volume and pressure overload. These peptides act as true hormones outside the heart and as paracrine or autocrine signals within the heart, exerting a wide spectrum of cardioprotective effects that extend to blood vessels, kidneys, adrenal glands, and the nervous system [3,4,5]. Although BNP shares many of the ANP effects, and both act via the same main receptor—NPR-A—some aspects of their physiology seem to be more complementary than redundant [6]. While BNP appears primarily to antagonize the adverse remodeling of the heart exposed to pressure overload stress (thereby reducing afterload), ANP appears to initiate pathways preventing or ameliorating hypertension and hypervolemia (thus aimed to reduce preload) [6,7]. This discrepancy has been clearly demonstrated in pro-BNP or pro-ANP knockout mice, with the former developing heart fibrosis without hypertension or cardiac hypertrophy [8], and the latter exhibiting hypertension and heart hypertrophy [9,10]. Lower plasma levels of BNP, as compared with ANP, under physiological conditions may contribute to these differences [11], given that transgenic mice overexpressing BNP (10- to 100-fold increase in plasma BNP) display lowered blood pressure.

Most researchers agree that BNPs are predominantly produced in the atria, with the minimal (~1000-fold lower) contribution of the ventricles. However, in severe congestive heart failure, the ventricles become the primary site of BNP synthesis and release [11]. Contrary to ANP, the principal mechanism of BNP release relies on rapid gene expression and peptide synthesis de novo rather than the secretion of BNPs stored in granules [11]. The biosynthesis pathway for BNPs reproduces the pattern documented for other peptide hormones, originating with an inactive precursor (preprohormone) undergoing intracellular modification to prohormone (pro-BNP), which is eventually cleaved to 32-aa BNP and 76-aa N-terminal fragment (NT-proBNP). NPs are produced and released in response to muscle stretch (indicating augmented cardiac mechanical stress) [12], and circulating neurohormones (endothelin 1, angiotensin II) [13]; furthermore, cytokines, like IL-1β and TNF-α, have been shown to promote BNP, but not ANP secretion [14]. A simplified algorithm of NP physiology is presented in Figure 1.

The physiological actions of NPs are mediated by the guanylyl cyclase-coupled receptors expressed in a wide array of cells. Three types of receptors for cardiac NPs have been found: NPR-A, NPR-B, and NPR-C. NPR-A (also called GC-A) is the principal receptor for both ANP and BNP, expressed in blood vessels, heart, adipose tissue, kidneys, lungs, adrenal glands, brain, and liver. NPR-B binds the C-type natriuretic peptide (not covered in this review), while NPR-C, expressed in endothelial cells, kidneys, and adipose tissue, functions as a clearance receptor for all three NPs, with higher binding affinity for ANP than BNP, which results in longer BNP half-life in plasma [3,4].

### 3.2. Local, Cardiac Actions of BNP

BNP acting via paracrine and autocrine pathways displays antihypertrophic, antifibrotic, and antiarrhythmic properties [4]. Although separating the direct antihypertrophic effects from the consequences of improved blood pressure control is challenging, the development of left ventricular hypertrophy *before* the onset of hypertension in a rat model of genetic BNP deletion provides substantial support for the causal role of BNP in this context [15]. In line with this, the plasma level of NT-proBNP was an independent predictor of left ventricular hypertrophy development in patients without heart failure [16]. Noteworthy, however, is the fact that Rubattu et al. [17] found the genetic variants of ANP and NPR-A, but not BNP, to be associated with left ventricular hypertrophy in a cohort of hypertensive patients.

Preventing cardiac fibrosis appears to be the most prominent action of BNP, as evidenced by studies on mice with a targeted disruption of the BNP gene, displaying focal fibrotic lesions in the ventricles, without concurrent hypertension or ventricular hypertrophy [18]. Antagonizing the signaling pathway of profibrotic factors, in particular norepinephrine, angiotensin II and TGF-β, provides a plausible underlying mechanism [4].

The antiarrhythmic effects of NPs may, at least in part, stem from previously discussed antihypertrophic and antifibrotic properties, given that the adverse cardiac remodeling provides the anatomical substrate for cardiac arrhythmias [19]. Furthermore, the immunomodulatory effects of BNP may play a role in this context. BNP was shown to suppress monocyte chemotaxis directly, via the NPR-A pathway [20]. In line with this, BNP levels have been found to predict the onset and progression of atrial fibrillation [21,22,23], and the risk of sudden cardiac death and ventricular arrhythmias [24]. Longitudinal changes in NT-proBNP were related to the risk of atrial and ventricular arrhythmias in the ARIC community-based cohort study [25].

### 3.3. Systemic Actions of BNP

Outside the heart, NPs contribute to blood pressure control and water–electrolyte homeostasis via multiple mechanisms. ANP and BNP directly reduce vascular smooth muscle tone and increase capillary permeability [26]. It is worth noting that data from genetically modified mice indicate that functional NPR-A—the principal receptor for ANP and BNP—is essential for the acute blood pressure-lowering action of ANP, whereas chronic control of blood pressure is not affected by NPR-A deletion in smooth muscle tissue [27]. Contrary to that, the lack of endothelial NPR-A produces hypertension and hypervolemia and abolishes ANP-induced increase in microvascular albumin permeability. Regarding human data, BNP infusion decreased mean arterial pressure and systemic vascular resistance in most studies in heart failure patients but failed to exert similar effects in healthy subjects (see [28] for a review).

The potent diuretic and natriuretic effects of ANP and BNP are believed to stem from an increase in glomerular filtration rate and filtration fraction (as the result of afferent arteriole dilation and efferent arteriole constriction), and reduction in Na^+^ reabsorption in the collecting duct [29]. The resultant disruption of the osmotic gradient in the kidney medulla complements the aforementioned effects, along with the suppression of the renin–angiotensin–aldosterone system at different levels. However, a recent study by Heinl et al. [30] demonstrated that NPR-A is highly expressed in glomerular podocytes, mesangial cells, vascular smooth muscle cells, and vascular endothelium, but not in the tubular compartment, thereby calling into question the direct effects of ANP and BNP on the renal tubules [31]. BNP infusion has been repeatedly shown to increase urine volume and Na^+^ excretion in both heart failure patients and healthy men. These effects were accompanied by no change in renal plasma flow, while glomerular filtration rate was typically increased in healthy subjects, but not in heart failure patients [28].

The dynamic, bi-directional crosstalk between cardiac NPs and the sympathetic nervous system has been postulated for decades [32]; however, some aspects remain largely unknown [33]. The stimulation of alpha-adrenergic pathways promotes ANP and BNP release, whereas the beta-adrenergic system carries the opposite effect [32]. In turn, ANP has been repeatedly shown to inhibit sympathetic activity targeting various circulatory areas, including renal vasculature in animal studies [34,35,36,37], whereas data from human studies are inconsistent. Furthermore, very few studies addressed the role of BNP in the modulation of sympathetic activity. Low-dose intravenous infusion of BNP reduced cardiac sympathetic nerve activity in both healthy controls and patients with chronic heart failure, while the reduction in renal sympathetic nerve activity was found for the higher BNP dosage in the heart failure group only [38]. Contrary to that, BNP did not inhibit cardiac sympathetic nerve activity in a sheep model [33].

BNP suppresses the renin–angiotensin–aldosterone system by direct inhibition of renin and aldosterone production [39,40,41], thereby contributing to the diuretic and natriuretic actions reviewed above.

## 4. BNP and NT-proBNP Norms in Children

The established cut-off values for BNP and NT-proBNP in adults are crucial for accurately diagnosing, monitoring, and determining the appropriate timing for intervention in congestive heart failure. However, interpreting these biomarkers in children requires a different approach due to the wide variation in normal levels throughout childhood and the different etiologies of heart failure in pediatric patients.

Several studies have demonstrated that NT-proBNP and BNP serum levels can vary significantly across different pediatric age groups (Table 1).

The most dynamic changes are observed during the first days of life, with higher values tending to decrease throughout the neonatal period. NT-proBNP levels in healthy neonates can range from 260 to 13,224 pg/mL, reflecting the adaptive changes related to the elimination of the placenta and subsequent redistribution of blood to the lungs. This leads to an increase in ventricular volume and pressure load, which probably stimulates the secretion of natriuretic hormones. This results in such high concentrations in the neonatal period [7]. In contrast, children aged between 1 month and 1 year typically exhibit NT-proBNP levels ranging from 5 to 1121 pg/mL [44]. Similar trends are observed in BNP values, which range from 48.4 pg/mL to 231.6 pg/mL during the first week of life, decreasing to less than 32.7 pg/mL in neonates older than two weeks [42]. Physiological BNP and NT-proBNP concentrations in children gradually reach typical adult values around the age of 18.

In general, there is no significant sex-related difference in BNP and NT-proBNP levels between genders. However, during the second decade of life, BNP levels are higher in girls than in boys, as described by Koch and Singer [42], but there are also reports that these values are higher in boys [45].

There is also no clear data on the influence of puberty on the concentration of natriuretic hormones. Kiess et al. prove that, during this period, there is a decrease in NT-proBNP in both sexes: in girls until stage 4 of the Tanner scale, and in boys until the end of puberty [45].

Numerous research trials have compared NT-proBNP levels in groups of healthy children of different ages with those in children with heart disease, establishing its role as a biomarker of heart failure. However, there is limited data on NT-proBNP levels in children with other conditions such as infections, obesity, or dehydration. Some authors suggest that NT-proBNP levels are more closely related to clinical deteriorations, such as shock, rather than to regular inflammatory diseases [43]. This issue is addressed in more detail in further parts of this review.

### Physical Exercise and NPs

NPs have been shown to increase significantly during forced and prolonged physical exercise like marathon running but also in soccer players, reaching peak concentrations immediately after the exercise and remaining increased for 72 h due to either myocardial injury or reduced renal clearance connected with temporal kidney function reduction following physical stress [46,47,48]. The mentioned studies are mostly based on adult patients, but a comparison of NT-proBNP increase in teenage and adult swimmers did not reveal significant differences according to age—in all the groups regardless of puberty status, the increase in NT-proBNP was observed after one hour of swimming [49]. Such conclusions may not apply to younger children, especially considering the endurance of the exercise required to increase NPs. However, it is important to include the preceding physical exercise anamnesis in the interpretation of NPs elevation in emergency situations in children without heart diseases.

Most of the data on NPs in children come from single-center studies; therefore, this topic requires further investigation. Current guidelines for using these biomarkers in pediatric populations are still lacking, highlighting the need for comprehensive, multicenter studies to establish standardized guidelines and improve clinical practice.

## 5. Clinical Applications of BNP and NT-proBNP in Children with Underlying Heart Diseases

In recent decades, extensive research on B-type natriuretic peptide (BNP) and N-terminal pro-BNP (NT-proBNP) within the pediatric population has provided substantial evidence of the utility of these biomarkers in various clinical settings. Natriuretic peptides (NPs) are now routinely utilized not only in pediatric cardiology units but also in neonatal intensive care units, general pediatrics, and emergency departments. Although their most common use is in the monitoring of heart failure in children with congenital heart disease or cardiomyopathy, the management of children post-cardiac surgery and the monitoring of the progression of pulmonary hypertension, these biomarkers are also valuable in the screening for heart disease (such as genetic dilated cardiomyopathies) in undiagnosed patients and in differentiating between cardiac and pulmonary disease prior to a comprehensive cardiological diagnostic process.

Regular monitoring of NP levels is recommended for all patients with cardiomyopathies, regardless of the underlying cause, as it has prognostic and therapeutic monitoring value. The recommendations indicate that increased NP levels are markers for decompensation and a tool for monitoring treatment effectiveness, but they do not specify precise cut-offs depending on age.

Despite the rapid fluctuations in NT-proBNP levels during the first days of life, numerous studies have been conducted to evaluate the clinical relevance of BNP/NT-proBNP assays in managing the neonatal patent arterial duct (PDA). Interestingly, BNP and NT-proBNP values were found to correlate with the volume of the shunt and end-diastolic volume [50,51,52].

The potential of BNP/NT-proBNP in monitoring children after cardiac surgery is significant, as according to some authors, mean BNP levels generally decrease within the first 24 h post-surgery in most neonates. However, an increase in BNP levels above the preoperative baseline at 24 h is associated with low cardiac output syndrome (LCOS) and prolonged mechanical ventilation, and can predict adverse outcomes such as death, unplanned operations, or cardiac transplants with a sensitivity of 80% and specificity of 90% [53].

In a specific group of pediatric patients with pulmonary hypertension, BNP/NT-proBNP levels have also proven valuable as monitoring markers. Elevated NT-proBNP levels correlate with higher WHO Functional Class (WHO-FC), lower six-minute walk distance (6MWD) z-scores, and lower tricuspid annular plane systolic excursion (TAPSE) z-scores. These elevated levels are also associated with an increased risk of mortality or the necessity for heart–lung transplantation. This highlights the prognostic significance of NT-proBNP in this patient population [54].

Heart failure due to acquired heart disease is less common in pediatric patients than in adults, and therefore, only a limited number of studies address this issue. The majority of these studies refer to Kawasaki disease (KD) or multi-system inflammatory syndrome in children (MIS-C), which are discussed in detail below. Myocarditis is another possible cause. The American Heart Association recommendations on myocarditis in children advise monitoring natriuretic peptides (NPs) as markers for potential HF as a complication of myocarditis but indicate that myocarditis is not a cause for increased NPs themselves. Akgul et al. [55] in their study on 62 patients, showed that BNP values were higher in patients with myocarditis requiring intensive care treatment than in those with a more benign disease [56].

The widespread application of BNP and NT-proBNP assays highlights the critical need for age-appropriate reference ranges to ensure accurate clinical interpretation.

## 6. Clinical Applications of BNP and NT-proBNP in Children without Underlying Heart Diseases

### 6.1. BNP and NT-proBNP in MIS-C and Kawasaki Disease

Kawasaki disease is an acute inflammatory disease that mostly affects preschool children of median age 2 years. The precise pathomechanism is not yet known but environmental (most probably infectious) factors in connection with host response dependent on genes and immune system play a role. KD is a small and medium-sized vessel vasculitis with multi-systemic manifestation, and its clinical diagnosis is based on the exclusion of infectious causes of a disease and the presence of a few days of fever along with mucocutaneous findings and lymphadenopathy. Cardiac involvement in Kawasaki disease (KD) can be extensive, affecting the pericardium, myocardium, endocardium, and coronary arteries, with common findings including tachycardia, systolic flow murmurs, and rare instances of pericardial tamponade or significant valvular dysfunction, primarily mitral regurgitation. Immunomodulatory treatment implemented in a timely manner not only cures the disease but also decreases the risk of lifelong lasting complications, such as coronary artery aneurysm (CAA)—the most common cardiac consequence of KD. Acute heart failure is a possible but rare manifestation and may be secondary to extensive myocardial infarction from thrombosis of aneurysms [57].

Multi-system inflammatory syndrome in children is an immune-mediated complication of primary SARS-CoV-2 infection that happens mostly in school-aged unvaccinated children. It clinically resembles Kawasaki disease, but distinguished molecular mechanisms have been identified. The differences are also expressed in clinical manifestations with more often gastrointestinal involvement as well as prominent cardiac involvement in MIS-C. Children with MIS-C more commonly suffer acute heart failure in the course of the disease while CAA is a possible but relatively rare complication. HF is acute and, in the majority of cases, reversible. A significant increase in NPs is noted in children with as well as without decreased left ventricular ejection fraction.

Belhadjer’s study on 35 MIS-C patients hospitalized at PICU was one of the first descriptions of severe cardiac involvement in MIS-C presenting as acute heart failure. This study revealed very high NPs among MIS-C children and it was a novelty in pediatric acute acquired inflammatory disorders. Dorio et al. [58] used this characteristic finding for further proteomic landscape investigation and not only proved that MIS-C and acute COVID-19 are two different diseases but also indicated a possible pathomechanism for MIS-C overlapping more with macrophage activation syndrome (MAS) and thrombotic microangiopathy (TMA) [59].

According to a meta-analysis of cardiac parameters in MIS-C, 60% of patients had elevated BNP levels, and 87% had elevated NT-proBNP levels [60]. The dynamics of NT-proBNP concentration during MIS-C depends on the duration of the fever with prominently higher values in the sixth day of the disease as compared to the third day [61].

Our previous study covers one of the largest cohorts concentrating on cardiac involvement in MIS-C [62]. We also observed a high proportion of patients with increased BNP levels, with a median value of 1070 pg/mL (264–5245), and an even larger percentage of children with elevated NT-proBNP levels, with a median value of 4744 pg/mL (1462–11,479). It is worth noting that less than 30% of these patients presented with contractility abnormalities and decreased left ventricular ejection fraction (LVEF), and approximately a quarter of them showed no echocardiographic abnormalities, suggesting that the increase in NPs may reflect other clinical issues rather than heart failure. Conversely, children whose initial echocardiographic results were classified as normal often exhibited a significant increase in LVEF on follow-up tests. This implies that either the sensitivity of the thresholds established for diagnosing heart failure was too low to detect the true impact of MIS-C, or that more subtle discrepancies depended on acute volume changes due to increased vascular permeability rather than actual heart failure in these cases.

Clinical manifestations of MIS-C and KD can be very similar, especially in younger children. NPs are listed as laboratory features distinguishing between these two entities. Table 2 presents chosen studies with the values of NPs found in children with MIS-C and KD.

Ashraf S. Harahsheh took an interesting approach and stratified patients with symptoms resembling KD or MIS-C by the status of SARS-CoV-2 infection, revealing significantly higher concentration of NPs, particularly NT-pro-BNP in children with SARS-CoV-2 exposure than in SARS-CoV-2 naive population [76]. Suchitra V Surve suggests that cardiac involvement (reflected as elevated NPs) is a good indicator of MIS-C among patients with acute symptoms and SARS-CoV-2 infection [77].

Acute heart failure is a less common complication of Kawasaki disease [78], and therefore not many pre-pandemic studies evaluated NPs in KD patients. With waves of MIS-C observed in 2020–2022, more attention has been given to NPs also in KD.

NT-pro-BNP, along with interleukin 17 (Il17), was proposed as a set of laboratory tests for differentiation between incomplete Kawasaki disease and other inflammatory conditions in children with fever; the cut-off value for NT-pro-BNP in such usage was established as 225.5 pg/dl [76]. Their meta-analysis indicated that patients with KD had higher BNP (52–142 pg/mL) as well as NT-proBNP (750–1511 pg/mL) levels than children with other febrile illnesses (BNP 4–60 and NT-proBNP 47–199 pg/mL), placing it among the useful markers of a disease but not good enough as a single KD diagnostic test [78].

If KD needs to be differentiated from other lymphadenopathies, the 18.3 pg/mL BNP threshold proved useful [79].

Feng et al. found that the concentration of BNP correlated with coronary artery lesion risk, but the values of BNP were relatively low (less than 300 µg/L) in both groups—with and without coronary artery involvement [80]. Bing et al. had similar findings [81]. It is worth noticing that these studies were conducted in the Asian population, and the results for other ethnic groups may not be applicable, similar to the Kobayashi score, which serves well only in Eastern parts of the globe. Another limitation is that children of younger ages particularly prone to CAL may be an independent factor impacting both CAL risk as well as physiologically higher NT-proBNP concentration (as already presented).

The meta-analyses of factors indicating the higher risk of first-line treatment failure in KD revealed that higher pro-BNP could serve as such an indicator along with some other laboratory measurements [82,83]. Similarly to the studies concentrating on the risk of CAL, the limitation of this approach is that it is based on few studies covering mainly Asian population patients.

Iwashima et al. noted a negative correlation between NT-pro-BNP and sodium concentration, hematocrite and albumins serum concentration in KD patients, implying its reflection of increased vascular permeability [84]. Interestingly, not only the consequences of systemic inflammation may lead to an increase in NPs in acute inflammatory diseases such as MIS-C or KD, but the infusion of high-dose intravenous immunoglobulins, which is a treatment of choice for both of them, can be a single factor leading to further increase in NT-pro-BNP (probably due to their impact on preload) [85].

### 6.2. BNP and NT-pro-BNP in Other Acute Conditions in Children without Underlying Heart Disease

#### 6.2.1. Sepsis and NPs

Patients with sepsis have significantly increased NP plasma concentrations. The increase is reported in patients without as well as with HF, but several studies aimed at elucidating the cut-off values that would differentiate a subgroup of pediatric patients with HF in the course of sepsis. In Lin’s study [86], NT-pro-BNP concentration values were increased in patients with sepsis as compared to the healthy control group, but particularly high in patients with sepsis and heart failure. The authors of this study claimed that NT-pro-BNP is a good tool for evaluating heart involvement and cardiac complications of sepsis. The cut-off value differentiating HF among patients with sepsis was established as 1268 ng/L. On the other hand, in Fried et al.’s study, only some but not all patients with sepsis and HF had distinguishingly high NPs, and there was a significant overlap between the NT-pro-BNP values of patients with sepsis and HF [87]. The small study sample was a major limitation of that study. Current studies have led to the conclusion that NPs have prognostic value in patients with sepsis but cannot be used to identify the cause of the shock (e.g., cardiogenic vs. septic shock) [1]. The majority of the studies reveal that the severity of the inflammatory response is reflected by the height of NPs concentration [88,89]. However, even the prognostic value of NPs needs to be treated with caution, as not all studies support such a claim [90]. Age-dependent factors may play a crucial role. Sun et al. proposed different cut-off values indicating increased risk of short-term mortality in pediatric sepsis depending on age [91]: 5000 ng/L for children aged less than 1 year, 4500 ng/L for 1–3 years old, 4100 ng/L for 4–6 years old, and 3800 ng/L for 7–18 years old children and teenagers.

A small number of patients showed that NT-pro-BNP in KD is higher than in sepsis and other inflammatory diseases (pre-MIS-C era) [68].

#### 6.2.2. Other Acute Diseases and NPs

NPs can rise in acute lung injury/acute respiratory distress syndrome, but in theory the concentrations in cardiogenic pulmonary edema should be higher. In adult patients, NPs serve as a good tool in the differentiation of the origins of pulmonary edema. An investigation of BNP as a potential marker for differentiation between acute lung injury/acute respiratory distress syndrome from cardiogenic pulmonary edema in children did lead to inconsistent, age-dependent and time-dependent results, making this measure less reliable than in adults [92]. Studies on bronchiolitis revealed that raised concentrations of BNP (less commonly investigated NP-pro-BNP) are correlated with more severe disease course, even without cardiac involvement, but in such situations BNP reaches few hundred pg/mL while, if complicated with acute heart failure, BNP reaches several thousand pg/mL [93]. A prominent increase in BNP (reaching nearly 1000 pg/mL) may also indicate congenital heart disease in a child with bronchiolitis [94]. Fluid imbalance during bronchiolitis may contribute to NPs elevation [95]. The authors of another study that supports the claim that, the more severe the increase in NPs, the higher risk of severe bronchiolitis course, suggest that this test could be used to stratify the disease severity risk in the emergency department [96].

Borensztajn et al., in their analysis of NPs in febrile children, indicated that the values of the NPs reflect the severity of a disease rather than its underlying cause, as there were no significant differences between children with viral or bacterial diseases. However, children with underlying heart conditions, as well as those hospitalized at PICU and with sepsis, had higher NP serum concentrations [97]. This supports Kim’s previous findings, who compared NT-pro-BNP in patients with cardiac disease, infectious disease, and non-cardiac and non-infectious disease, where patients with underlying cardiological problems had the highest NP levels, but for the others the increase in NPs reflected the disease severity. Higher NT-pro-BNP values were found among patients requiring mechanical ventilation, inotropic support, or with altered mental status. A positive correlation with procalcitonin was noted [43].

NT-pro-BNP was proposed as a useful biomarker to perceive increased risk of cardiac events and complications in patients with chronic kidney disease, especially in advanced stages of renal impairment [98,99,100]. It seems, however, that there are limitations in the utility of NPs as it correlates with the glomerular filtration rate in children as compared to adults [101].

## 7. Summary—Limitations and Challenges of NT-pro-BNP Testing in Children

When interpreting BNP or NT-proBNP levels in children with acute diseases but without underlying heart problems, healthcare providers should consider the context of the patient’s condition. Elevated BNP or NT-pro-BNP levels in these cases may reflect the body’s response to stress rather than primary cardiac dysfunction. It is essential to differentiate between BNP or NT-pro-BNP elevations due to acute illnesses and those indicating underlying heart issues to avoid unnecessary interventions or misdiagnoses. Although B-type natriuretic peptide (BNP) levels can be elevated in children with infectious diseases, significantly higher levels are observed in those with congenital or acquired heart disease. This difference provides a valuable tool for distinguishing between cardiac and non-cardiac causes of acute stress in pediatric patients [102]. Based on the current literature on different clinical scenarios, a conclusion appears, which is that the values of thousands of pg/mL rather than a few hundred pg/mL reflect cardiac involvement.

By incorporating NT-pro-BNP measurements alongside clinical evaluation and relevant diagnostic tests, healthcare providers can better understand the overall hemodynamic status of the child and tailor their management accordingly.

Further research is needed to better understand the role of NT-pro-BNP in children with acute diseases and to establish clear cut-off values and guidelines for its interpretation in this population. By considering the broader context of a patient’s clinical presentation, healthcare providers can effectively utilize NT-pro-BNP testing to optimize care for children with acute illnesses.

## 8. Conclusions

NT-pro-BNP appears to be more useful than BNP in clinical settings.

By incorporating NT-pro-BNP measurements alongside clinical evaluation and relevant diagnostic tests, healthcare providers can better understand the overall hemodynamic status of the child and tailor their management accordingly.

Monitoring NT-pro-BNP levels in febrile children without underlying heart diseases can help healthcare providers assess the impact of acute illness on cardiac function and fluid status. Significantly increased NP levels may serve as a marker of increased risk of a severe course in acute febrile diseases, bronchiolitis, and bronchopneumonia in children, as well as potential indicators of first-line treatment failure and coronary artery lesion (CAL) formation in Kawasaki disease.

Based on the current literature on different clinical scenarios, it appears that NT-pro-BNP values in the thousands of pg/mL, rather than a few hundred pg/mL, reflect cardiac involvement.

Therefore, in prioritizing febrile children who require cardiological consultation and echocardiography, a significant increase in NT-pro-BNP levels into the thousands should guide the decision. If the levels are in the hundreds, we tend to consider inflammatory disease as the cause of the NP elevation.

Children with congenital heart defects and cardiomyopathies should have regular NT-pro-BNP activity tests to monitor for heart failure.

## Figures and Tables

**Figure 1 ijms-25-08781-f001:**
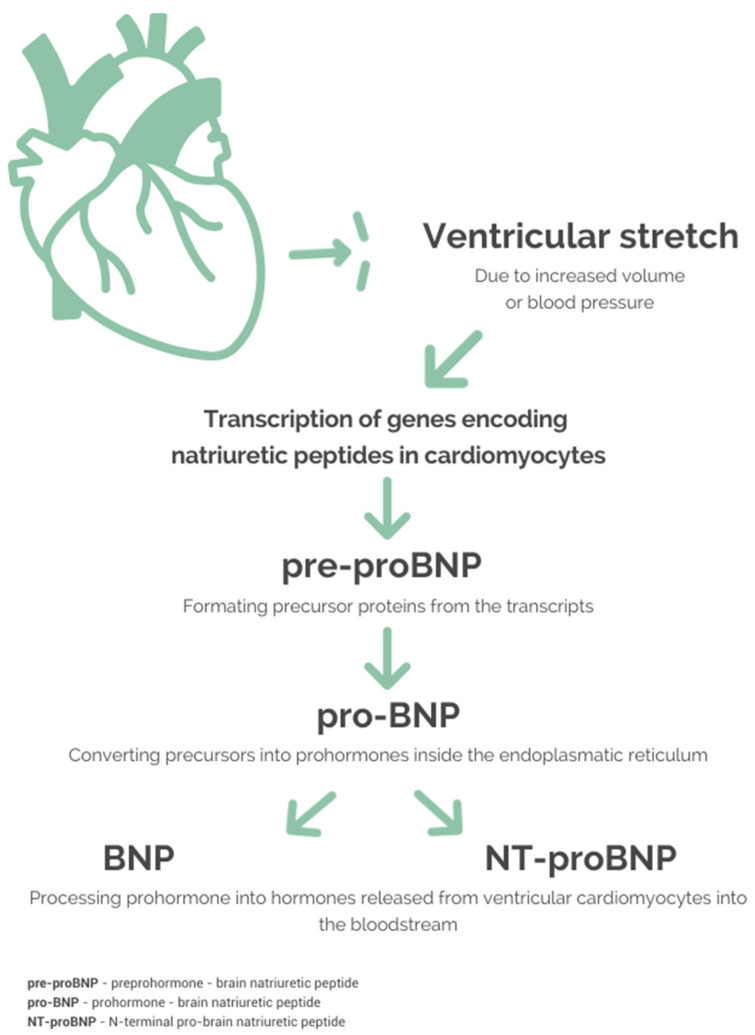
Physiology of BNP and NT-proBNP production.

**Table 1 ijms-25-08781-t001:** BNP and NT-proBNP norms in different age groups.

Authors	Year of Publication	Age Group	BNP (pg/mL) *	NT-proBNP (pg/mL) *	Link
A Koch, H Singer Heart [42]	2003	0–1 day	231.6		https://pubmed.ncbi.nlm.nih.gov/12860862/ (URL accessed on 9 June 2024)
A Koch, H Singer Heart [42]	2003	4–6 days	48.4		https://pubmed.ncbi.nlm.nih.gov/12860862/ (URL accessed on 9 June 2024)
Hyun Su Kim and Hee Joung Choi [43]	2020	0 to 17.5 years		280	https://pubmed.ncbi.nlm.nih.gov/32102709/ (URL accessed on 9 June 2024)
Amiram Nir et al. [44]	2008	0–2 days		3183	https://pubmed.ncbi.nlm.nih.gov/18600369/ (URL accessed on 9 June 2024)
Amiram Nir et al. [44]	2008	3–11 days		2210	https://pubmed.ncbi.nlm.nih.gov/18600369/ (URL accessed on 9 June 2024)
Amiram Nir et al. [44]	2008	>1 mo to 1 yr		141	https://pubmed.ncbi.nlm.nih.gov/18600369/ (URL accessed on 9 June 2024)
Amiram Nir et al. [44]	2008	1 yr to 2 yr		129	https://pubmed.ncbi.nlm.nih.gov/18600369/ (URL accessed on 9 June 2024)
Amiram Nir et al. [44]	2008	2 yr to 6 yr		70	https://pubmed.ncbi.nlm.nih.gov/18600369/ (URL accessed on 9 June 2024)
Amiram Nir et al. [44]	2008	6 yr to 14 yr		52	https://pubmed.ncbi.nlm.nih.gov/18600369/ (URL accessed on 9 June 2024)
Amiram Nir et al. [44]	2008	14 yr to 18 yr		34	https://pubmed.ncbi.nlm.nih.gov/18600369/ (URL accessed on 9 June 2024)

* Median or mean values are provided following the source studies. BNP: B-Type Natriuretic Peptide; NT-proBNP: N-Terminal pro B-Type Natriuretic Peptide; pg/mL: Picograms per Milliliter.

**Table 2 ijms-25-08781-t002:** NT-proBNP and BNP values in Kawasaki disease and multi-system inflammatory syndrome in children.

Authors	Year of Publication	Number of Participants	Medical Condition	NT-proBNP (pg/mL)	BNP (pg/mL)	Comments	Link
Zhang R et al. * [63]	2020	100	KD	350		differentiating from (higher than in) other febrile illness	https://pubmed.ncbi.nlm.nih.gov/33378024/ (URL accessed on 11 June 2024)
Ganguly et al. [64]	2022	72	KD	914.91			https://www.ncbi.nlm.nih.gov/pmc/articles/PMC8652561/ (URL accessed on 12 June 2024)
Ganguly et al. [64]	2022	71	MIS-C	9141.16		good in predicting MIS-C	https://www.ncbi.nlm.nih.gov/pmc/articles/PMC8652561/ (URL accessed on 12 June 2024)
Song HB et al. * [65]	2020	95	KD	1310		higher in patients with CAL	https://pubmed.ncbi.nlm.nih.gov/31931930/ (URL accessed on 11 June 2024)
Cai WJ et al. * [66]	2022	139	KD	506.31		higher in patients with CAL	https://pubmed.ncbi.nlm.nih.gov/36596080/ (URL accessed on 12 June 2024)
Desjardins et al. [67]	2020	127	KD	2029		higher in children with systolic disfunction	https://pubmed.ncbi.nlm.nih.gov/32172336/ (URL accessed on 12 June 2024)
Banerjee P et al. [68]	2023	40	KD	914.91		higher than in other febrile illness	https://pubmed.ncbi.nlm.nih.gov/37551875/ (URL accessed on 12 June 2024)
Joung J et al. * [69]	2022	896	KD	1828		higher in IVIG non-responders with cut-off value 1561 pg/mL and higher in patients with CAL with cut-off value 789.0 pg/mL	https://www.ncbi.nlm.nih.gov/pmc/articles/PMC9354345/ (URL accessed on 13 June 2024)
Bilal M et al. * [70]	2020	500	KD			higher in patients with CAL	https://www.ncbi.nlm.nih.gov/pmc/articles/PMC7405981/ (URL accessed on 13 June 2024)
Jung JH et al. [71]	2023	378	KD		59	higher than in other febrile illness, with BNP > 270 pg/mL higher risk of CAL in KD	https://pubmed.ncbi.nlm.nih.gov/37478221/ (URL accessed on 14 June 2024)
Otar Yener G et al. [72]	2022	154	MIS-C	1108		higher than in KD	https://www.ncbi.nlm.nih.gov/pmc/articles/PMC8421714/ (URL accessed on 14 June 2024)
Otar Yener G et al. [72]	2022	59	KD	55			https://www.ncbi.nlm.nih.gov/pmc/articles/PMC8421714/ (URL accessed on 14 June 2024)
Belhadjer et al. [57]	2020	35	MIS-C		4256		https://www.ahajournals.org/doi/10.1161/CIRCULATIONAHA.120.048360 (URL accessed on 14 June 2024)
Whittaker et al. [73]	2020	58	MIS-C	788		NT-proBNP levels may be helpful in predicting progression of disease	https://jamanetwork.com/journals/jama/fullarticle/2767209 (URL accessed on 15 June 2024)
Bichali et al. [74]	2023	31	MIS-C	32,933		measured at 6 days from first symptoms	https://doi.org/10.1007/s12519-022-00681-8 (URL accessed on 11 June 2024)
Ludwikowska et al. [61]	2023	498	MIS-C			different values across age groups for BNP but not for NT-proBNP	https://www.ncbi.nlm.nih.gov/pmc/articles/PMC10215748/ (URL accessed on 10 June 2024)
Valverde et al. [75]	2021	286	MIS-C				https://www.ahajournals.org/doi/full/10.1161/CIRCULATIONAHA.120.050065 (URL accessed on 15 June 2024)

* The median values were provided for the subgroups of patients with one disease; therefore, an average of a range value is provided. BNP: B-type natriuretic peptide; CAL: coronary artery lesion; KD: Kawasaki disease; MIS-C: multi-system inflammatory syndrome in children; NT-proBNP: N-terminal pro-B-type natriuretic peptide; pg/mL: picograms per milliliter; μg/L: micrograms per liter; ng/L: nanograms per liter.

## Data Availability

This is a review article; therefore, all the data were extracted from the source articles included in the reference list.

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
