# Peer review of "Clinical Significance of B-Type Natriuretic Peptide and N-Terminal Pro-B-Type Natriuretic Peptide in Pediatric Patients: Insights into Their Utility in the Presence or Absence of Pre-Existing Heart Conditions"

_ijms, 2024, doi:10.3390/ijms25168781_

Round 1
Reviewer 1 Report
Comments and Suggestions for Authors
- some references should be included in the text body as numbers, not only as author's names (ex: reference 9, reference 62, reference 86);
- reference 89 appears before reference 46;
- reference 90 is included before reference 58;
- please insert the abbreviation for CAA
- it would be worthwhile to include, if possible, the BNP and NT-proBNP values in evolution, in those patients with critical condition and death, due to Kawasaki disease, MIS-C, sepsis, acute respiratory distress syndrome, with and without underlying heart disease.
Author Response
Dear Reviewer,
We were very pleased to receive such a favorable review. We appreciate the effort you’ve put into improving our work with your valuable comments. We believe that incorporating your suggestions will enhance the value and utility of the manuscript. We’ve made the corrections according to your suggestions—please find our responses to your comments below:
- Some references should be included in the text body as numbers, not only as author's names (e.g., reference 9, reference 62, reference 86) - corrected as revised.
- Reference 89 appears before reference 46 - corrected as revised.
- Reference 90 is included before reference 58 - corrected as revised.
- Please insert the abbreviation for CAA - corrected as revised.
- It would be worthwhile to include, if possible, the BNP and NT-proBNP values in evolution in those patients with critical conditions and death due to Kawasaki disease, MIS-C, sepsis, acute respiratory distress syndrome, with and without underlying heart disease. - Thank you for the interesting idea. We agree that it would be valuable to address this issue, but the differences in the methodology of the revised manuscripts could lead to misleading conclusions. We believe that the values gathered in the text and tables provide the best overview of the threshold of BNP and NT-proBNP values concerning the clinical problem. For your information, we plan to conduct a study including patients with various clinical problems to define the cut-off values for natriuretic peptides among the pediatric population and its changes over time, considering the differences related to age and the severity of the condition, as well as the existence of underlying cardiac disease.
Thank you again for your valuable feedback.
Best regards,
Authors
Reviewer 2 Report
Comments and Suggestions for Authors
Ludwikowska KM et al. manuscript contains an elaborate literature review about the current understanding of BNP and NT-proBNP utility in children in the presence or absence of pre-existing heart disease conditions, taking into account clinical contexts such as fever, infection, or inflammation.
The authors, showing a good knowledge of the subject matter, appropriately discusses the findings according to current data and existing literature.
Title and Abstract
● The title is appropriate and the abstract is sufficiently informative and is consistent with the content of the paper.
Introduction
● The introduction clearly defines the main aspects of the topic being investigated and tells the reader what the manuscript will be about. The authors briefly describe the background and present the aim of the review in the context of what is already known about the topic.
Physiology of BNP...
● Very well-written and well-articulated section showing the authors' accurate knowledge of the topic.
Minor commenti: please consider trying to be a little more concise in order to make it easier for other researchers to reproduce and verify your work. Moreover, please consider using shorter sentences to improve readability. In fact, some sentences are a little bit too long (see sentence from row 75 to row 80).
● Please consider using for in-text citations only bibliographical reference number and not accompany them with the name of the first author (see row 75, row 82, row 88 etc).
● Two minor comments: reference n^ 9 is missing in the text and please consider to close
parenthesis on row 2 on page 3.
BNP and NT-proBNP norms in children
This section reviews literature data on serum levels of BPN and NT-pro-BNP in different groups of pediatric patient and highlights the limitations of the data they examined.
However, for correctness of the information, I would suggest to add a sentence on the reasons why they choose to examine only the listed publications.
Clinical applications of BNP and NT-proBNP in children with underlying heart disease
Tyis section is well really interesting, complete and exhaustive. Minor comments:
● Row 291: I would use “ according to a metanalysis” instead of “ according to the metanalysis”.
● Table 2 : check the patients number in Cai WJ publication. They are 139.
● Row 347 : Typo (“preambiliy”).
Summary
In this section the authors show main conclusions of the study, hoping for further future studies in this area.
Please consider to add a reference to the sentence on Row 375 “was established as 1268 ng/L.”
Comments on the Quality of English Language
English style is fine and require minor editing
Author Response
Dear Reviewer,
We were very pleased to receive such a favorable review. We appreciate the effort you’ve put into improving our work with your valuable comments. We believe that incorporating your suggestions will enhance the value and utility of the manuscript. We’ve made the corrections according to your suggestions—please find our responses to your comments below:
- Minor commenti: please consider trying to be a little more concise in order to make it easier for other researchers to reproduce and verify your work. Moreover, please consider using shorter sentences to improve readability. In fact, some sentences are a little bit too long (see sentence from row 75 to row 80). - Thank you for this suggestion. We have made several changes regarding the grammar and consistency of our text, both in the mentioned sentence and in several other places. You can track the changes in the manuscript; the lines you referred to have been revised as follows: “Since the discovery of an “extremely powerful” natriuretic substance in rat atrial myocytes in 1981 [2], atrial natriuretic peptide (ANP) and brain natriuretic peptide (BNP) have emerged as critical regulators of the physiological response to cardiac volume and pressure overload. These peptides act as true hormones outside the heart and as paracrine or autocrine signals within the heart, exerting a wide spectrum of cardioprotective effects that extend to blood vessels, kidneys, adrenal glands, and the nervous system.’
- Please consider using for in-text citations only bibliographical reference number and not accompany them with the name of the first author (see row 75, row 82, row 88 etc). - corrected as revised.
- Two minor comments: reference n^ 9 is missing in the text and please consider to close parenthesis on row 2 on page 3. - corrected as revised.
BNP and NT-proBNP norms in children
This section reviews literature data on serum levels of BPN and NT-pro-BNP in different groups of pediatric patient and highlights the limitations of the data they examined.
However, for correctness of the information, I would suggest to add a sentence on the reasons why they choose to examine only the listed publications. - The most current available publications on the role of BNP and NT-proBNP in the management of various clinical conditions were reviewed. For the sake of clarity, we have decided to add Chapter 2 on Methodology. This chapter will detail the study design, data collection methods, and analysis techniques used in our research. We believe this addition will enhance the transparency and reproducibility of our findings. Thank you for this valuable suggestion.
Clinical applications of BNP and NT-proBNP in children with underlying heart disease
Tyis section is well really interesting, complete and exhaustive. Minor comments:
- Row 291: I would use “ according to a metanalysis” instead of “ according to the metanalysis”. - corrected as revised.
- Table 2 : check the patients number in Cai WJ publication. They are 139. - To our surprise, after reanalyzing this publication, we discovered a discrepancy between the data in the abstract and the content of the article. The abstract states that the study involved 139 patients with KD, whereas in subsection 2.2, the authors state that "A total of 155 children were included in this study (...)", which is the number we reported in our manuscript. After summing up the number of all participants, the number 139 seems to be correct, so in accordance with your suggestion, we have made this correction. (link to the publication: https://www.ncbi.nlm.nih.gov/pmc/articles/PMC9803503/)
- Row 347 : Typo (“preambiliy”). - corrected as revised.
Summary
Please consider to add a reference to the sentence on Row 375 “was established as 1268 ng/L.” - corrected as revised.
Thank you again for your valuable feedback.
Best regards,
Authors
Reviewer 3 Report
Comments and Suggestions for Authors
Dear authors, I have read with interest the paper "Assessing the Clinical Significance of BNP and NT-proBNP in Pediatric Patients: Insights into their Utility in the Presence or Absence of Pre-existing Heart Conditions."
Despite the topic being of interest to the reader of IJMS, several issues should be addressed
1) The paper lacks a Figure. Can I suggest adding one figure on the mechanism of actions of NPs
2) No mention of the use of NPs in myocarditis and genetic dilated cardiomyopathies. Please add.
3) I don't understand the need for a section on Physical exercise and NPs. Please consider deleting it
4) Add a conclusion sections
5) An algorithm on the use of NPs in clinical practice should be added
Comments on the Quality of English Language
The paper should be extensively revised by a native English speaker
Author Response
Dear Reviewer,
We are very grateful for the review we received. We appreciate the effort you’ve put into improving our work with your valuable comments. We believe that incorporating your suggestions will enhance the value and utility of the manuscript. We’ve made some corrections according to your suggestions—please find our responses to your comments below:
1) The paper lacks a Figure. Can I suggest adding one figure on the mechanism of actions of NPs - Thank you for this valid suggestion. We have created a figure illustrating the basics of the physiology of natriuretic peptides. We hope you will find it relevant and complementary to the content of our manuscript. Additionally, it seems that you might not have had the chance to review the graphical abstract we prepared to complement our article—we believe it effectively supplements the issues discussed.
2) No mention of the use of NPs in myocarditis and genetic dilated cardiomyopathies. Please add. - According to your suggestion, we have added a subsection on the value of natriuretic peptides in the mentioned diseases.
3) I don't understand the need for a section on Physical exercise and NPs. Please consider deleting it - While working on the manuscript, we analyzed various articles that addressed the values of natriuretic peptides in conditions associated with and without heart disease. Many of these articles indicated a significant correlation between increased NP values and intense physical exercise. We believe that from a clinical standpoint, knowledge of this cause of elevated NP levels is very important and can prevent taking unnecessary diagnostic steps. Therefore, we decided to retain this subsection in the content of our manuscript – it was shifted to the “BNP and NT-proBNP norms in children” section.
4) Add a conclusion sections – We added a conclusion section as Chapter 8 at the end of the article.
5) An algorithm on the use of NPs in clinical practice should be added - We agree with the excellent idea that creating such an algorithm would be helpful in differentiating conditions with elevated NP levels. However, since we are not relying on our own research, the methodologies of the studies we cite differ from each other, and we do not have the appropriate systems and authorizations to create clinical guidelines, we have refrained from formulating such an algorithm. We deeply believe that the study we plan to conduct, aimed at defining the cut-off values of NP in pediatric diseases, will make a significant contribution to the formulation of such guidelines in the future.
Thank you again for your valuable feedback.
Best regards,
Authors
Round 2
Reviewer 3 Report
Comments and Suggestions for Authors
No futher comments